# LLMCBench: Benchmarking Large Language Model Compression for Efficient Deployment

**Ge Yang**[1], **Changyi He**[1], **Jinyang Guo**[1,*] **Jianyu Wu**[1], **Yifu Ding**[1],
**Aishan Liu**[1], **Haotong Qin**[2], **Pengliang Ji**[3], **Xianglong Liu**[1]
[1] Beihang University    [2] ETH Zurich    [3] Carnegie Mellon University

## Abstract

Although large language models (LLMs) have demonstrated their strong intelligence ability, the high demand for computation and storage hinders their practical application. To this end, many model compression techniques are proposed to increase the efficiency of LLMs. However, current researches only validate their methods on limited models, datasets, metrics, etc, and still lack a comprehensive evaluation under more general scenarios. So it is still a question of which model compression approach we should use under a specific case. To mitigate this gap, we present the Large Language Model Compression Benchmark (LLMCBench), a rigorously designed benchmark with an in-depth analysis for LLM compression algorithms. We first analyze the actual model production requirements and carefully design evaluation tracks and metrics. Then, we conduct extensive experiments and comparison using multiple mainstream LLM compression approaches. Finally, we perform an in-depth analysis based on the evaluation and provide useful insight for LLM compression design. We hope our LLMCBench can contribute insightful suggestions for LLM compression algorithm design and serve as a foundation for future research. Our code is available at https://github.com/AboveParadise/LLMCBench.

## 1 Introduction

Recently, large language models (LLMs) have attracted increasing attention because of their strong intelligence ability. While it achieves excellent performance, the huge computation and storage burden hinders the practical usage of these LLMs. To solve this problem, many model compression techniques specifically designed for efficient LLMs have been proposed in recent years, including sparsification [6, 35], quantization [46, 33], knowledge distillation [8], and so on.

Among these compression technologies, sparsification and quantization are two mainstream approaches for LLM compression, and most LLM compression methods recently proposed belong to these two categories. However, existing LLM compression works are still far away from practical usage due to two main challenges:

***Challenge-1*: Performance evaluation scope is limited.** The emergence of large language models is less than two years, and this is still an active research area now. Following this trend, new types of LLMs have surged quickly in recent years. It causes a problem that the current LLM compression researches often use different types of LLMs for evaluation, which cannot form a fair comparison between different methods. For example, the classic quantization method SmoothQuant [46] uses OPT [49], BLOOM [21], and GLM [5] as the base model for evaluation, while the latter approach OmniQuant [33] utilizes LLaMA [38] for evaluation. The evaluation protocol can be very different between different methods. Moreover, even the base model performance is different in current works. For example, the perplexity of LLaMA-7B in sparsity method LLM-Pruner [27] is 12.62,

---

*Corresponding Author: Jinyang Guo <jinyangguo@buaa.edu.cn>

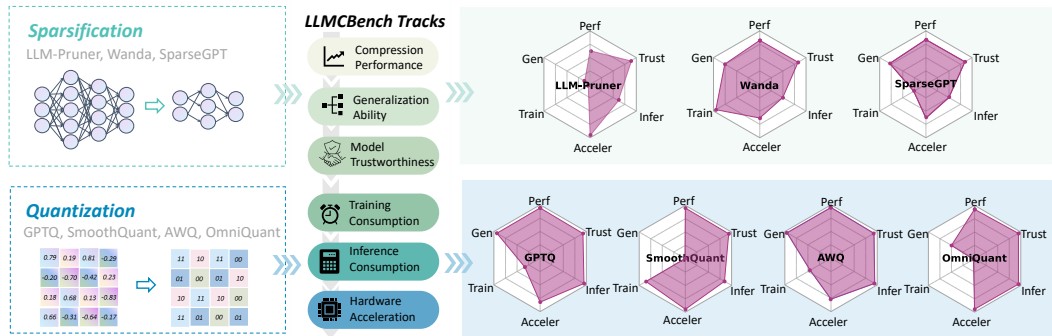

Figure 1: Overview of our LLMCBench.

while this value is 5.68 in OmniQuant. Furthermore, current researches use various datasets in performance comparison. All aforementioned problems hinder a comprehensive evaluation of recent LLM compression works, posing the question of which LLM compression algorithm is effective under certain scenarios.

*Challenge-2*: **Efficiency evaluation metric remains theoretical.** Most of the LLM compression approaches in the literature choose computation complexity or model storage as the efficiency metric. However, it still lacks a comprehensive evaluation on broader efficiency metrics such as practical acceleration, GPU memory reduction, and so on. Moreover, the resource consumption of the compression process is often ignored in the current evaluation protocol. Further, as the compressed LLM needs to be used in practical scenarios, model trustworthiness is also an important aspect of the compression algorithm, which is not considered in the current research [25]. So the question of which LLM compression is suitable for real-world model production still remains.

In this paper, we present **L**arge **L**anguage **M**odel **C**ompression **Bench**mark (**LLMCBench**), the first benchmark to provide a comprehensive evaluation on current LLM compression algorithms. Starting from real-world model production requirements, we carefully design 6 tracks to fairly compare featured sparsity and quantization methods including LLM-Pruner [27], Wanda [35], SparseGPT [6], GPTQ [7], SmoothQuant [46], AWQ [23], and OmniQuant [33]. We chose these methods mainly because sparsity and quantization are two mainstream LLM compression techniques, and these approaches are widely-used and open-sourced[2]. We benchmark these 6 representative algorithms on 11 datasets, 18 network architectures, and 3 deployment platforms. Based on the extensive experiments, we provide an in-depth analysis on LLM compression algorithms and offer useful insights and suggestions for LLM compression algorithm design.

In summary, we construct LLMCBench, the first benchmark to comprehensively evaluate various LLM compression algorithms. It provides a systematic comparison from brand-new perspectives for the practical production of lightweight LLMs. Based on the extensive evaluation results, we conduct an in-depth analysis and provide insightful suggestions. We hope our LLMCBench can push LLM compression algorithms toward practical usage.

## 2 Background

### 2.1 Large Language Models

The emergence of large language models (LLMs) has become a milestone in the field of natural language processing. With the development of LLMs, decoder-based LLM has become the mainstream structure. For example, [30] proposed the GPT model to stack multiple transformer decoder blocks. Meta released LLaMA [38] based on an improved transformer architecture, which is further extended to LLaMA2 [39] and LLaMA3. In this paper, we choose 6 LLMs including LLaMA [38], LLaMA2 [39], LLaMA3, Vicuna [50], OPT [49], and ChatGLM [5] for evaluation to construct our LLMCBench, which are the most representative LLMs in the current research. Moreover, multimodal LLMs are advancing by integrating text and vision [43, 52, 51], enabling models to process and generate content across different modalities, enhancing applications like image captioning and visual question answering.

---

[2]Each method received more than 500 stars on GitHub in one year.

## 2.2 Model Compression

To reduce the massive computation and parameter burden of LLMs, many model compression methods were proposed in recent years. These works mainly focus on sparsity [6, 35, 45, 17, 2, 12, 11, 10, 15, 14, 41], quantization [46, 33, 13, 26], and knowledge distillation [8]. Among these approaches, sparsity and quantization are the most popular two techniques[3]. Thus, we choose these two techniques to construct our LLMCBench.

**Model sparsification.** Model sparsification aims to remove unimportant weights or activations to construct a sparse model to reduce the parameter and computation of LLMs [19, 36, 18, 22]. It can be roughly categorized into unstructured sparsity, structured 2:4 sparsity, and structured sparsity. Unstructured sparsity removes individual weights irregularly to obtain sparse models. Structured sparsity removes the entire channel for structured matrix computation. Structured 2:4 sparsity removes two weights in each four-weight block. To comprehensively benchmark sparsity algorithms, we choose the most representative methods in each category for evaluation, i.e., SparseGPT [6] and Wanda [35] for unstructured and structured 2:4 sparsities, and LLM-Pruner [27] for structured sparsity.

**Model quantization.** Model quantization aims to quantize the weight or activation in LLMs using lower bit numbers to reduce computation and parameters [42, 4, 47]. It can be roughly categorized into post-training quantization (PTQ) and quantization-aware training (QAT). Considering the high training cost of LLMs, the PTQ paradigm is more popular in the current research. To this end, we choose the four most representative quantization methods in this category in our LLMCBench, which includes GPTQ [7], SmoothQuant [46], AWQ [23], and OmniQuant [33].

## 2.3 Challenges of LLM Compression

LLM compression methods have attracted increasing attention since 2023. However, as the LLM compression algorithms have emerged quickly in recent years, several challenges still remain in the current research. First, the performance evaluation protocols are different and limited. Different compression methods may select different baseline LLMs and datasets to evaluate their approach. This evaluation protocol may cause unfair comparison and also lacks a comprehensive comparison on specific abilities of LLMs, posing the question of which LLM compression method is more effective in a specific scenario. Second, the efficiency evaluation metrics are still theoretical in current research. Most LLM methods only report the #MACs or #parameters or acceleration after compression but do not consider other important factors in real-world production and deployment like training consumption and acceleration on different libraries etc. Moreover, the compressed LLMs are expected to be used in real-world applications. So the model trustworthiness after compression is also a crucial aspect for LLM compression algorithms, which is not considered in existing evaluation. Under this background, we construct our LLMCBench for comprehensive LLM compression algorithm evaluation.

# 3 LLMCBench: Tracks and Metrics

In this section, we introduce the competition tracks and metrics in our LLMCBench, which consists of six tracks. A higher score of the metric indicates better performance. For better readability, we have multiplied the theoretical score by 100.

## 3.1 Track 1: Compression Performance

Current LLM compression methods only compare the performance on several specific datasets but lack comprehensive evaluation on different abilities. In our LLMCBench, we divide the mainstream evaluation dataset into two main abilities: knowledge ability and inference ability. The knowledge ability indicates whether the LLM knows the world, while the inference ability indicates whether the LLM can reason based on its knowledge.

---

[3]None of the distillation methods received more than 300 stars on GitHub, compared with over 500 stars for representative sparsity/quantization approaches.

To quantitatively reflect the compression performance of different algorithms, we define the following overall metric (OM) across all models and datasets:

$$\text{OM}_{\text{perf}} = \sqrt{\frac{1}{N} \sum_{i=1}^{N} \mathbb{E}\left(\frac{A^c_{\text{ability}_i}}{A_{\text{ability}_i}}\right)^2}, \tag{1}$$

where $A^c_{\text{ability}_i}$ and $A_{\text{ability}_i}$ are the accuracy of the compressed model and pre-trained model for the $i$th abilities, respectively. $N$ is the number of abilities. $\mathbb{E}(\cdot)$ is the mean operation. In this case, we calculate the mean accuracy over the datasets and models.

We use the quadratic mean to unify all track metrics, preventing outliers from skewing the results and ensuring more accurate measurements.

### 3.2 Track 2: Generalization Ability

We also evaluate the generalization ability of different LLM compression methods in our LLMCBench. An effective LLM compression algorithm should be effective for various model types and sizes, but existing researches only choose specific LLM families and sizes for evaluation. We design the overall metric of this track as follows:

$$\text{OM}_{\text{gen}} = \sqrt{\frac{1}{N} \sum_{i=1}^{N} \mathbb{E}\left(\frac{A^c_{\text{mod}_i}}{A_{\text{mod}_i}}\right)^2}, \tag{2}$$

where $A^c_{\text{mod}_i}$ and $A_{\text{mod}_i}$ are the accuracy of the compressed model and pre-trained model for the $i$th model type, respectively. $N$ denotes the number of model types. In this track, we calculate the mean value of different model sizes under one model type in the mean operation $\mathbb{E}$.

### 3.3 Track 3: Training Consumption

The third track in our LLMCBench is training consumption. It is intuitive that an effective LLM compression algorithm should require small resources to finish the compression process, but existing compression approaches lack comprehensive evaluation from this aspect. In this track, we evaluate the training consumption from two perspectives including time consumption and GPU memory usage. Time consumption indicates the time cost of LLM algorithms to finish the compression, while GPU memory usage evaluates the maximum required memory of each LLM compression method. Similar to the previous two tracks, we design the following overall metric for this track:

$$\text{OM}_{\text{train}} = \sqrt{\frac{1}{2}\left(\mathbb{E}\left(\frac{T^{\text{max}}_{\text{train}}}{T_{\text{train}}}\right)^2 + \mathbb{E}\left(\frac{M^{\text{max}}_{\text{train}}}{M_{\text{train}}}\right)^2\right)}, \tag{3}$$

where $T^{\text{max}}_{\text{train}}$ and $M^{\text{max}}_{\text{train}}$ are the maximum training time and GPU memory in all the evaluated methods for corresponding models and datasets. In this track, we calculate the mean value of training time and memory consumption over all models and datasets. The terms $T^{\text{max}}_{\text{train}}$ and $M^{\text{max}}_{\text{train}}$ are used for normalization to ensure we have a higher overall metric for better performance.

### 3.4 Track 4: Inference Consumption

Inference consumption is one of the most critical aspects of LLM compression algorithms for efficiency evaluation. However, current researches still lack systemic evaluation on this perspective. In our LLMCBench, we benchmark inference consumption from three main aspects: computation complexity, model size, and GPU memory consumption in the inference stage. Similar to other tracks, we design the overall metric of this track as:

$$\text{OM}_{\text{inf}} = \sqrt{\frac{1}{3}\left(\mathbb{E}\left(\frac{M_{\text{inf}}}{M^c_{\text{inf}}}\right)^2 + \mathbb{E}\left(\frac{S_{\text{inf}}}{S^c_{\text{inf}}}\right)^2 + \mathbb{E}\left(\frac{F_{\text{inf}}}{F^c_{\text{inf}}}\right)^2\right)}, \tag{4}$$

where $M_{\text{inf}}$, $S_{\text{inf}}$, and $F_{\text{inf}}$ are GPU memory, model size, and the number of MACs for pre-trained LLM at inference stage, respectively. $M^c_{\text{inf}}$, $S^c_{\text{inf}}$, and $F^c_{\text{inf}}$ are those of compressed LLM at inference stage, respectively. In the mean operation $\mathbb{E}$, we calculate the mean value over all models and datasets for the corresponding metric.

### 3.5 Track 5: Hardware Acceleration

Hardware acceleration is another important aspect of LLM compression algorithms for efficiency evaluation [9]. The implementation details in current compression methods often have a large impact on this aspect. Even the same compression method may have different acceleration performances on different libraries. Existing LLM compression approaches seldom extensively compare this important aspect, making the acceleration performance remain theoretical. Similar to previous tracks, we define the following overall metric for this track:

$$\text{OM}_{\text{hard}} = \sqrt{\frac{1}{N} \sum_{i=1}^{N} \mathbb{E}\left(\frac{V_{\text{lib}_i}^c}{V_{\text{lib}_i}}\right)^2}, \tag{5}$$

where $V_{\text{lib}_i}$ and $V_{\text{lib}_i}^c$ are the token generation speed of pre-trained and compressed models on the $i$th library, respectively. In this track, we take the average value over all models and datasets on the $i$th library in the mean operation $\mathbb{E}$. $N$ is the number of libraries we used in this track.

### 3.6 Track 6: Trustworthiness

The compressed LLMs need to be deployed in real-world scenarios. So model trustworthiness of the deployed LLMs is also a critical aspect [34] to avoid negative social impact. However, current compression methods do not include the trustworthiness evaluation when comparing performance. In our LLMCBench, we also evaluate the LLM compression algorithms from the trustworthiness perspective. Specifically, we divide model trustworthiness into robustness and truthfulness. Robustness refers to the ability of LLMs to properly handle malicious adversarial attack text and out-of-distribution input, while truthfulness indicates whether an LLM can output correct facts under the interference of noise, erroneous information, bias, etc. Similar to other tracks, we design the following metric for this track:

$$\text{OM}_{\text{trust}} = \sqrt{\frac{1}{2}\left(\mathbb{E}\left(\frac{A_{\text{rob}}^c}{A_{\text{rob}}}\right)^2 + \mathbb{E}\left(\frac{A_{\text{tru}}^c}{A_{\text{tru}}}\right)^2\right)}, \tag{6}$$

where $A_{\text{rob}}$ and $A_{\text{tru}}$ are the accuracy for the pre-trained LLMs on the robustness and trustfulness task, respectively. $A_{\text{rob}}^c$ and $A_{\text{tru}}^c$ are those for the compressed LLMs, respectively. We also take the average over all models and datasets for the mean operation $\mathbb{E}$.

## 4 LLMCBench Implementation

**Implementation details.** We implemented LLMCBench using PyTorch and conducted our experiments on Nvidia A800 GPUs. Given pre-trained LLMs, we use different LLM compression algorithms to compress the model to obtain the compressed LLM. For LLM-Pruner [27], we set the sparsity ratio as 50% for all tracks. For Wanda [35] and SparseGPT [6], we evaluate both 50% unstructured sparsity and structured 2:4 sparsity. As GPTQ [7] and AWQ [23] are weight-only quantization methods, we use 8bit for the weight (W8A16). For SmoothQuant [46] and OmniQuant [33], we set both weight and activation as 8bit (W8A8). One exception is we use W4A16 for GPTQ and AWQ and use W4A4 for SmoothQuant and OmniQuant in Track 2 as there is no significant performance difference for 8-bit quantization in this track. All the hyperparameters are the same as the open-sourced code from the original approaches.

**Evaluation protocal.** For track 1, we adopt commonly used MMLU [16], Arc-eacy [3], and Arc-challenge [3] to evaluate the knowledge ability of LLMs. For inference ability, we choose six datasets including Hellaswag [48], PIQA [1], WinoGrande [32], QNLI [31], MNLI [44], and WikiText2 [28] for evaluation. Regarding model selection, we choose two popular decoder-based LLMs: LLaMA2-7B [39] and LLaMA3-8B for evaluation. For track 2, we extensively choose four model families including LLaMA [38], Vicuna [50], OPT [49], and ChatGLM [5], and also include different model sizes ranging from 6B to 70B. We evaluate the performance of compressed model on WikiText2. For track 3 and track 4, we use LLaMA2-7B and LLaMA3-8B on WikiText2 for evaluation, as they are widely used in many compression methods [46, 35]. For track 5, we choose three representative deployment libraries: TensorRT-LLM [29], vLLM [20], and MLC-LLM [37] to evaluate the acceleration of different algorithms. We categorize the algorithms into

Table 1: Compression performance of different methods. LMA2 and LMA3 refer to LLaMA2-7B and LLaMA3-8B, respectively. H.S. means HellaSwag.

| Method | Model | Sparsity /#Bits | Knowledge ability | | | H.S. | PIQA | Inference ability | QNLI | MNLI | Wiki↓ | $OM_{ka}$ | $OM_{ia}$ | $OM_{perf}$ |
|---|---|---|---|---|---|---|---|---|---|---|---|---|---|---|
| | | | MMLU | ARC-c | ARC-e | | | Wino | | | | | | |
| **Sparsity** | | | | | | | | | | | | | | |
| Dense | LMA2 | 0 | 40.52 | 46.33 | 74.58 | 75.98 | 79.11 | 69.06 | 50.53 | 44.31 | 5.12 | 100 | 100 | 100 |
| | LMA3 | 0 | 61.38 | 53.50 | 77.74 | 79.12 | 80.69 | 73.24 | 50.86 | 63.48 | 5.54 | | | |
| LLM-Pruner | LMA2 | 50% | 24.15 | 27.47 | 46.52 | 47.76 | 68.44 | 54.14 | 49.45 | 34.33 | 20.66 | 60.51 | 75.85 | 68.61 |
| | LMA3 | 50% | 29.90 | 32.17 | 55.09 | 55.93 | 69.70 | 62.51 | 50.60 | 40.71 | 14.22 | | | |
| Wanda | LMA2 | 50% | 29.67 | 42.75 | 69.07 | 70.78 | 76.66 | 68.90 | 50.67 | 35.28 | 6.46 | 83.25 | 90.19 | 86.79 |
| | LMA3 | 50% | 40.59 | 44.97 | 68.18 | 68.23 | 76.01 | 70.17 | 50.60 | 54.57 | 8.61 | | | |
| Wanda | LMA2 | 2:4 | 23.63 | 32.25 | 58.46 | 55.11 | 71.71 | 62.43 | 50.64 | 35.12 | 6.51 | 62.53 | 78.78 | 71.12 |
| | LMA3 | 2:4 | 27.57 | 28.84 | 50.04 | 47.86 | 66.10 | 59.83 | 50.60 | 32.44 | 19.98 | | | |
| SparseGPT | LMA2 | 50% | 34.62 | 42.24 | 67.89 | 71.04 | 76.44 | 69.69 | 50.62 | 35.16 | 6.51 | 85.10 | 91.29 | 88.25 |
| | LMA3 | 50% | 48.33 | 42.15 | 65.70 | 71.66 | 76.71 | 70.32 | 50.60 | 54.96 | 7.55 | | | |
| SparseGPT | LMA2 | 2:4 | 25.76 | 33.62 | 60.23 | 58.68 | 72.36 | 66.14 | 50.61 | 36.05 | 10.28 | 67.53 | 81.29 | 74.73 |
| | LMA3 | 2:4 | 28.27 | 33.87 | 57.15 | 56.02 | 68.28 | 63.69 | 50.60 | 42.50 | 10.96 | | | |
| **Quantization** | | | | | | | | | | | | | | |
| Full Prec. | LMA2 | FP16 | 40.52 | 46.33 | 74.58 | 75.98 | 79.11 | 69.06 | 50.53 | 44.31 | 5.12 | 100 | 100 | 100 |
| | LMA3 | FP16 | 61.38 | 53.50 | 77.74 | 79.12 | 80.69 | 73.24 | 50.86 | 63.48 | 5.54 | | | |
| GPTQ | LMA2 | INT8 | 40.77 | 46.25 | 74.33 | 76.00 | 79.11 | 68.90 | 50.62 | 39.53 | 6.88 | 99.97 | 97.17 | 98.58 |
| | LMA3 | INT8 | 61.36 | 53.41 | 77.69 | 79.06 | 80.63 | 72.85 | 50.77 | 63.44 | 5.54 | | | |
| SmoothQuant | LMA2 | INT8 | 39.02 | 44.28 | 73.36 | 74.41 | 78.18 | 66.93 | 50.22 | 38.53 | 5.53 | 97.50 | 96.55 | 97.03 |
| | LMA3 | INT8 | 58.30 | 51.96 | 79.67 | 78.13 | 79.54 | 72.61 | 51.40 | 62.90 | 6.28 | | | |
| AWQ | LMA2 | INT8 | 40.90 | 46.16 | 74.41 | 75.98 | 79.05 | 69.22 | 50.64 | 38.86 | 5.12 | 99.89 | 98.89 | 99.39 |
| | LMA3 | INT8 | 61.22 | 53.22 | 77.57 | 79.15 | 80.59 | 72.45 | 50.46 | 63.43 | 5.54 | | | |
| OmniQuant | LMA2 | INT8 | 40.32 | 45.65 | 74.75 | 75.94 | 79.00 | 69.22 | 50.55 | 43.59 | 5.12 | 99.21 | 99.63 | 99.42 |
| | LMA3 | INT8 | 61.19 | 52.13 | 77.61 | 79.23 | 80.52 | 72.61 | 50.73 | 62.56 | 5.55 | | | |

structured/structured 2:4/unstructured sparsity, and INT8/INT4 quantization, as they show similar acceleration performance. We evaluate these compression paradigms using tokens per second as the metric. For track 6, we choose AdvGLUE [40] to evaluate the robustness and use TruthfulQA [24] for truthfulness.

## 5 Evaluation and Analysis

### 5.1 Track 1: Compression Performance

**Quantization offers better overall performance.** Table 1 presents the results we evaluated in track 1. Quantization approaches have better overall performance than sparsity methods when compressing LLMs. For example, the overall metric score $OM_{perf}$ is smaller than 90 for most sparsity methods, while this metric score is over 95 for quantization approaches. This indicates quantization is more suitable to preserve LLM performance after compression.

**Sparsity is better for inference ability, while quantization is better for knowledge ability.** We also calculate the overall metric for knowledge ability $OM_{ka}$ and inference ability $OM_{ia}$. Sparsity approaches often have higher overall inference ability, while quantization methods prone to preserve knowledge ability of LLMs. This indicates that we should use sparsity methods to compress LLMs if we focus more on their inference ability, and use quantization methods if preserving their knowledge capability is more critical.

### 5.2 Track 2: Generalization Ability

**Weight-only quantization methods have good generalization ability under lower bit.** The weight-only quantization methods GPTQ and AWQ have better generalization ability, which achieve over 95 overall metrics. This is because LLM is more sensitive to activation quantization.

**SmoothQuant is less general.** As SmoothQuant involves activation quantization, it is less general to different models. This may be because SmoothQuant aims to deal with outliers, but outliers are different in different models.

**Most approaches cannot generalize well on ChatGLM2.** All evaluated methods cannot perform well on ChatGLM2 except weight-only quantization approaches. Therefore, we need to specifically design compression methods if we need to deploy this model.

Table 2: Generalization ability performance of different LLM compression methods.

| Model | Dense | LLM-Pruner | Wanda | SparseGPT | GPTQ | SmoothQuant | AWQ | Omniquant |
|---|---|---|---|---|---|---|---|---|
| LLaMA-7B | 5.68 | 19.20 | 7.09 | 6.73 | 6.61 | 380.77 | 5.78 | 11.26 |
| LLaMA-13B | 5.09 | 14.15 | 6.03 | 5.85 | 5.20 | 552.8 | 5.19 | 10.86 |
| LLaMA-30B | 4.10 | 9.86 | 5.18 | 5.07 | 4.25 | 1057.91 | 4.20 | 10.63 |
| LLaMA-65B | 3.53 | 8.34 | 4.55 | 4.37 | 3.76 | 890.32 | 3.61 | 9.17 |
| LLaMA2-7B | 5.12 | 18.43 | 6.46 | 6.51 | 5.25 | 1887.53 | 5.23 | 14.26 |
| LLaMA2-13B | 4.57 | 14.10 | 5.47 | 5.34 | 4.66 | 403.44 | 4.65 | 12.29 |
| LLaMA2-70B | 3.12 | 6.34 | 3.91 | 3.81 | 3.31 | 1306.59 | 3.21 | 9604.32 |
| LLaMA3-8B | 5.54 | 15.35 | 8.61 | 7.55 | 5.75 | 799.70 | 6.14 | 12735.95 |
| LLaMA3-70B | 2.59 | 8.40 | 5.01 | 4.92 | 4.71 | 274.00 | 3.06 | 37026.54 |
| Vicuna-7B | 6.33 | 19.11 | 7.95 | 7.90 | 6.50 | 2636.98 | 6.51 | 87.39 |
| Vicuna-13B | 5.57 | 15.99 | 6.63 | 6.44 | 5.66 | 494.89 | 5.65 | 60.22 |
| OPT-1.3B | 14.62 | 124.01 | 18.41 | 17.55 | 16.41 | 1412.51 | 14.92 | 98.6 |
| OPT-2.7B | 12.47 | 163.81 | 14.22 | 13.46 | 12.81 | 8749.80 | 12.70 | 360.26 |
| OPT-6.7B | 10.86 | 119.49 | 11.98 | 11.60 | 11.05 | 21492.23 | 10.96 | 12.24 |
| OPT-13B | 10.13 | 113.89 | 11.93 | 11.15 | 10.22 | 13176.12 | 10.29 | 11.65 |
| OPT-30B | 9.56 | 76.00 | 10.03 | 9.77 | 9.59 | 12765.02 | 9.61 | 10.31 |
| ChatGLM2-6B | 105.58 | 43499.38 | 3916.7 | 2534.85 | 122.97 | 5887.32 | 128.58 | 3624.92 |
| ChatGLM3-6B | 6.21 | 301.05 | 20.58 | 33.86 | 6.34 | 1175.5 | 6.4 | 494.41 |
| $OM_{gen}$ | 100 | 28.89 | 76.41 | 79.06 | 93.80 | 0.82 | 96.13 | 48.51 |

## 5.3 Track 3: Training Consumption

**Wanda requires the least training resources.** The results for track 3 are shown in Fig 2. The sparsity method Wanda requires the least training resources among these evaluated approaches. It has around 43 overall metric scores. On the other hand, the quantization method OmniQuant requires the most training resources. This is mainly because the compression time for OmniQuant is long.

**Learning is the bottleneck.** The compression methods requiring a learning process often have higher training consumption. For example, OmniQuant requires more than 300 A800 GPU minutes to finish the compression, as the retraining/learning process is time-consuming. Therefore, if we need fast compression speed, we need to choose compression methods without learning.

**SmoothQuant and AWQ require less GPU memory.** SmoothQuant and AWQ require less memory, while LLM-Pruner requires the highest one. This may be because LLM-Pruner and OmniQuant need to retrain the model, which takes more memory. Although Wanda, SparseGPT, and GPTQ do not require retraining, they need to calculate the sparsity/quantization metric based on the activation, which also takes more memory. Therefore, if the GPU memory is limited in the compression process, we can choose SmoothQuant or AWQ for compression.

## 5.4 Track 4: Inference Consumption

**Quantization generally has less inference consumption.** The results for track 4 are shown in Table 3. Quantization approaches often have higher overall metrics for inference consumption. This may be because quantization uses lower bits to represent full-precision numbers. Therefore, the GPU memory and model size will be reduced in the inference stage. On the other hand, although sparsity methods set unimportant weights/neurons to zero, they still need to be stored in the memory.

**LLM-Pruner is the best among sparsity methods.** LLM-Pruner is the structured sparsity method, while the others are unstructured or structured 2:4 sparsity. So the entire structure can be directly

removed for LLM-Pruner, while other sparsity methods fail to achieve this due to memory and cache issues. Therefore, we can choose structured sparsity methods if we want better inference consumption performance without special implementation.

**Quantization methods have similar inference consumption.** Although using different quantization techniques, the quantized models from different algorithms have similar inference consumption except for SmoothQuant, as SmoothQuant does not support real quantization deployment in their open-sourced code. However, we believe we can use other deployment libraries for real quantization.

Table 3: Inference consumption of different LLM compression methods in our LLMCBench.

| Method | Model | Sparsity/#Bits | GPU Memory | Model Size | #MACs | OM$_{inf}$ |
|---|---|---|---|---|---|---|
| **Sparsity** | | | | | | |
| Dense | LMA2 | 0 | 22.96G | 12.55G | 0.85T | 100 |
| | LMA3 | 0 | 25.35G | 14.96G | 0.97T | |
| LLM-Pruner | LMA2 | 50% | 13.50G | 6.75G | 0.51T | 161.86 |
| | LMA3 | 50% | 18.50G | 9.97G | 0.62T | |
| Wanda | LMA2 | 50% | 22.96G | 12.55G | 0.43T | 134.76 |
| | LMA3 | 50% | 25.35G | 14.96G | 0.57T | |
| Wanda | LMA2 | 2:4 | 22.96G | 12.55G | 0.43T | 134.76 |
| | LMA3 | 2:4 | 25.35G | 14.96G | 0.57T | |
| SparseGPT | LMA2 | 50% | 22.96G | 12.55G | 0.43T | 134.76 |
| | LMA3 | 50% | 25.35G | 14.96G | 0.57T | |
| SparseGPT | LMA2 | 2:4 | 22.96G | 12.55G | 0.43T | 134.76 |
| | LMA3 | 2:4 | 25.35G | 14.96G | 0.57T | |
| **Quantization** | | | | | | |
| Full-Precision | LMA2 | FP16 | 22.96G | 12.55G | 0.85T | 100 |
| | LMA3 | FP16 | 25.35G | 14.96G | 0.97T | |
| GPTQ | LMA2 | INT8 | 15.16G | 6.67G | 0.23T | 245.91 |
| | LMA3 | INT8 | 17.03G | 8.62G | 0.29T | |
| SmoothQuant | LMA2 | INT8 | 23.62G | 12.55G | 0.23T | 220.58 |
| | LMA3 | INT8 | 25.02G | 14.96G | 0.29T | |
| AWQ | LMA2 | INT8 | 15.15G | 6.71G | 0.23T | 245.11 |
| | LMA3 | INT8 | 17.72G | 8.66G | 0.29T | |
| OmniQuant | LMA2 | INT8 | 15.13G | 6.53G | 0.23T | 246.34 |
| | LMA3 | INT8 | 17.19G | 8.61G | 0.29T | |

Therefore, we can choose the most suitable quantization method to achieve the target inference consumption when compressing LLMs.

## 5.5 Track 5: Hardware Acceleration

**INT4 quantization has the best acceleration performance.** Fig 3 shows the hardware acceleration for track 5. The dark results represent testing on LLaMA2, and the light results represent testing on LLaMA3. (T) represents TensorRT-LLM, (V) represents vLLM, and (M) represents MLC-LLM. INT4 quantization can achieve promising speedup under various deployment libraries and achieves the highest overall metric under this track.

**Structured sparsity $\approx$ INT8 quantization.** Structured sparsity and INT8 quantization have similar overall metrics for this track, which indicates that structured sparsity and INT8 quantization can achieve similar speedup under different libraries.

**Structured 2:4 sparsity is not well-supported.** For structured 2:4 sparsity, only TensorRT-LLM can achieve acceleration. This may be because vLLM and MLC-LLM do not support this sparsity paradigm. Therefore, we can use TensorRT-LLM for deployment on this sparsity type and should put more effort into this sparsity paradigm.

## 5.6 Track 6: Trustworthiness

**Quantization brings better trustworthiness.** From the results, quantization methods provide better trustworthiness than sparsity approaches. $OM_{trust}$ are over 95 for quantization, while these numbers are below 95 for sparsity methods.

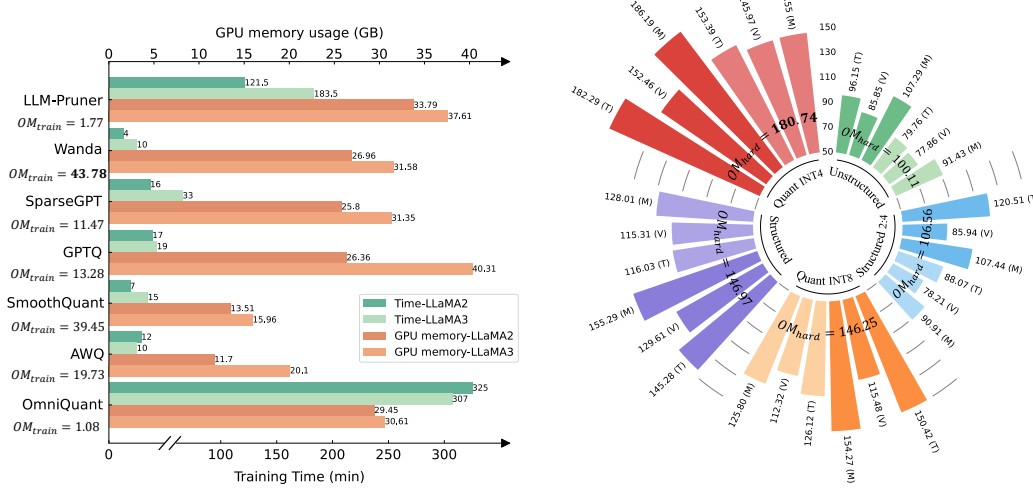

Figure 2: Training time and memory usage consumption.

Figure 3: Hardware acceleration of different types of LLM compression methods.

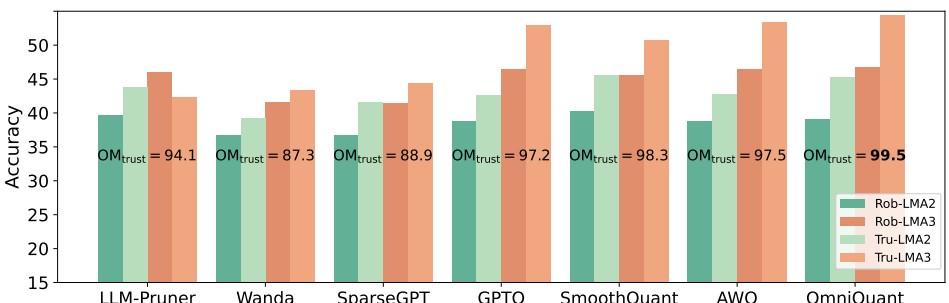

Figure 4: Trustworthiness of different LLM compression methods in our LLMCBench.

**Better compression performance $\neq$ better trustworthiness.** Different from the compression performance track, LLM-Pruner achieves the best $\mathrm{OM_{trust}}$ across all sparsity methods, which indicates better compression performance does not guarantee better trustworthiness. For example, LLM-Pruner achieves 94.09 overall metrics on this track, while the unstructured sparsity methods only have less than 90 overall metrics.

**Weight-activation quantization brings better trustworthiness.** Compared with weight-only quantization (i.e., GPTQ and AWQ), the weight-activation quantization paradigm (SmoothQuant and OmniQuant) has a higher overall metric. Therefore, it is beneficial to use weight-activation quantization for the trustworthiness consideration.

## 6 Discussion

From the evaluation of our LLMCBench, we have several conclusions: (1) Based on the current library and hardware development, quantization is more suitable for LLM compression because of better performance and hardware support. Considering the performance drop and acceleration, weight-only quantization like AWQ performs better than weight-activation quantization. (2) Weight-activation quantization like SmoothQuant is better in terms of inference efficiency (inference consumption and hardware acceleration). (3) Sparsity generally has better training efficiency. However, its hardware/library support is not well constructed in the current stage. It still requires further development in this area to achieve better compression performance.

# 7 Conclusion

In this paper, we presented a Large Language Model Compression Benchmark (LLMCBench) to systemically evaluate the LLM compression algorithms. Based on the evaluation results, we also provide an in-depth analysis to guide the further design of LLM compression approaches. We hope our LLMCBench can contribute insightful suggestions and serve as a foundation for future research.

One limitation of our LLMCBench is that we only choose the seven most representative approaches. We will include more LLM compression algorithms, such as LLM KV cache compression, in our future work. We will also introduce more tracks and datasets, such as coding datasets and mathematical datasets, to conduct more comprehensive tests on the compressed LLMs. Our LLMCBench aims to evaluate LLM compression algorithms for practical usage. So it does not have negative social impact.

## Acknowledgements

This work was supported by the Beijing Municipal Science and Technology Project (No. Z231100010323002), and the National Natural Science Foundation of China (Nos. 62306025, 92367204).

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
