# OpenReview forum: "LLMCBench: Benchmarking Large Language Model Compression for Efficient Deployment"
_NeurIPS.cc/2024/Datasets_and_Benchmarks_Track — NeurIPS 2024 Track Datasets and Benchmarks Spotlight_

### Official Review · Reviewer_jkFp · 2024-07-12
**An innovative benchmark for LLM compression methods**

**Rating:** 8
**Confidence:** 4
**Clarity:** yes

**Review:**

The contribution of this paper is a comprehensive assessment of the accuracy and efficiency of LLM compression methods, analysing a wide range of critical aspects. The authors present a thorough analysis of their proposed evaluation metrics as well as experimental results. The article is well structured and uses many charts to clearly present the results. With the help of LLMCBench, more compression approaches with excellent performance can emerge in the field of LLM compression. This article also has some shortcomings. The design of the sixth track is too simple, and the performance on the TruthfulQA and advGLUE does not necessarily represent the trustworthiness of the model. Some detailed experimental setups are not fully described.

**Strengths:**

1. This article is highly innovative. Most existing benchmarks evaluate various aspects of the performance of LLMs, while this paper considers testing different LLM compression methods.
2. The benchmark evaluates a wide range of compression techniques across multiple models, datasets, and platforms, providing a thorough analysis.
3. A large number of experimental results are presented in a reasonable way, which is convenient for readers to read.

**Additional Feedback:**

In the generalization ability experiments, do the LLMs you tested cover LLMs of all architectures?

Overall, I think the current state of the paper is pretty strong. I think the main thing that could strengthen this paper is the experiments on ANLI for trustworthiness evaluation.

**Correctness:**

This paper takes the performance of the full-precision model as a reference when designing metrics, which appears to be sound.

**Documentation:**

Some experimental details are not fully explained in this paper, such as the number of shots used in the experiment. Moreover, this article does not have open source code.

**Limitations:**

See Opportunities for improvement.

**Opportunities For Improvement:**

1. It would be better to also include the result on ANLI [1] for trustworthiness evaluation.
2. While the paper provides an overview of the implementation, more detailed information on the specific configurations and hyperparameters used for each compression method could enhance reproducibility.
3. In figure 1, it is better to also include overall metric for each method.

References:

[1] Nie, Yixin, et al. "Adversarial NLI: A new benchmark for natural language understanding." *arXiv preprint arXiv:1910.14599* (2019).

**Relation To Prior Work:**

yes

**Summary And Contributions:**

The high computational and storage demands of large language models (LLMs) hinder their practical application despite their strong intelligence capabilities. Recently, many model compression techniques have been proposed, but their evaluations are often limited to specific models, datasets, and metrics, lacking a comprehensive analysis under general scenarios. To address this, this paper presents a comprehensive benchmark for evaluating the effectiveness of various LLM compression techniques.This paper proposes six tracks from the perspectives of accuracy and efficiency, comprehensively evaluates various LLM compression algorithms, and provides profound insights.

---

> ### Author Rebuttal · Authors · 2024-08-16
>
> ****Q5:**** In the generalization ability experiments, do the LLMs you tested cover LLMs of all architectures?
>
> ****A5:**** In our LLMCBench, we tested 18 LLMs, including LLaMA, ChatGLM, etc., covering the current mainstream LLM architectures. The LLMs we evaluated range from the smallest 1.3B to the largest 70B, which also covers most LLM sizes. We will also evaluate the more recent MoE structure in our future work.

---

> > ### Comment · Reviewer_jkFp · 2024-08-24
> >
> > Thanks for the detailed response, and my initial concerns are well addressed. I think this paper is a greate contribution to the community. So I recommend to accept it.

---

> ### Author Rebuttal · Authors · 2024-08-16
>
> ****Q4:**** Some experimental details are not fully explained in this paper, such as the number of shots used in the experiment. Moreover, this article does not have open source code.
>
> ****A4:**** Sorry for the confusion. We use zero shot in all experiments and we will clarify this in our final version. We will also release the code after the paper is accepted.

---

> ### Author Rebuttal · Authors · 2024-08-16
>
> ****Q3:**** In figure 1, it is better to also include overall metric for each method.
>
> ****A3:**** Thanks for the suggestion! We will add the overall metric in figure 1 in our final version.

---

> ### Author Rebuttal · Authors · 2024-08-16
>
> ****Q2:**** While the paper provides an overview of the implementation, more detailed information on the specific configurations and hyperparameters used for each compression method could enhance reproducibility.
>
> ****A2:**** Sorry for the confusion. We follow the open-sourced code and use the same configuration and hyperparameters as the original methods for fair comparison. We will clarify this in our final version.

---

> ### Author Rebuttal · Authors · 2024-08-16
>
> We thank you for your professional and insightful comments. Our response to your comments is as below:
>
> ****Q1:**** It would be better to also include the result on ANLI for trustworthiness evaluation.
>
> ****A1:**** Thanks for your suggestion! We follow your suggestion and conduct the experiment on ANLI. The results are shown in the Table below. From the table, we have similar observations as the experiments in our main paper. We will add the results on ANLI in our final version.
>
> |      Models     | Dense/FP16 | LLM-Pruner | Wanda 50% | Wanda 2:4 | SparseGPT 50% | SparseGPT 2:4 | GPTQ  | AWQ   | OmniQuant | SmoothQuant |
> | --------- | ---------- | ---------- | --------- | --------- | ------------- | ------------- | ----- | ----- | --------- | ----------- |
> | LLaMA2-7B | 36.85      | 32.86      | 32.88     | 33.44     | 32.69         | 33.17         | 37.41 | 37.09 | 36.97     | 35.90        |
> | LLaMA3-8B | 35.56      | 35.57      | 34.11     | 33.41     | 33.59         | 33.41         | 35.60  | 35.58 | 35.83     | 34.51       |

---

### Official Review · Reviewer_bNyA · 2024-07-18
**Nice benchmark. Experiments could be better.**

**Rating:** 8
**Confidence:** 4
**Clarity:** The writting is clear.

**Review:**

LLMCBench solves the problem of difficulty in comparing various LLM compression algorithms by proposing a unified set of evaluation paradigms. Unlike existing benchmarks, the evaluation results in this paper are more helpful for deployment in practical applications. The advantages of this article can be found in the **Strength** section. Nevertheless, this article only focuses on open source models and does not consider closed source models of the GPT series.

**Strengths:**

1. The metrics designed are simple but effective, which measure the performance of different compression algorithms, and compare the advantages and disadvantages of pruning and quantization algorithms horizontally.
2. Overall the paper is well-organized and clearly written, making it easier for readers to understand.
3. The author uses the newly launched LLaMA3 model as the test object, adding some novelty to the article.

**Additional Feedback:**

N/A

**Correctness:**

The evaluation methods and experiment design are appropriate and performed correctly.

**Documentation:**

No GitHub repo link provided.

**Ethics:**

No specific ethical concerns found.

**Limitations:**

Please see the **Opportunities For Improvement** section.

**Opportunities For Improvement:**

1. I think the author should provide more experimental details, such as the number of shots used in the dataset, the calculation method of inference speed, etc.

2. In Table 3, can the semi-structured sparsity of Wanda and SparseGPT lead to #MAC reduction?

3. In your hardware acceleration results, why is semi-structured sparsity significantly accelerated in TensorRT-LLM, but not in the other two deployment platforms?

**Relation To Prior Work:**

This article could benefit from highlighting the differences between LLMCBench and other LLM evaluation benchmarks.

**Summary And Contributions:**

This paper proposes LLMCBench, a benchmark for evaluating large language models (LLMs) compression approaches with six features: compression performance, generalization ability, training consumption, hardware acceleration, and trustworthiness. It covers 7 LLM compression approaches and 18 LLMs, which is a great contribution. This paper also provides a detailed analysis of the experimental results and puts forward some ideas.

---

> ### Author Rebuttal · Authors · 2024-08-16
>
> ****Q6:**** No GitHub repo link provided.
>
> ****A6:**** Thanks for your suggestion! We will release the code after the paper is accepted.

---

> > ### Comment · Reviewer_bNyA · 2024-08-24
> > **Great rebuttal**
> >
> > Thanks for the efforts from the authors. My concerns are addressed in the rebuttal. I think this paper has its merit to the community. Therefore, I vote for acceptance of this paper and increase my rating to clear accept.

---

> ### Author Rebuttal · Authors · 2024-08-16
>
> ****Q5:**** This article could benefit from highlighting the differences between LLMCBench and other LLM evaluation benchmarks.
>
> ****A5:**** Thanks for your suggestion! Current benchmarks such as [R2] focus on evaluating the performance of language models, while others like [R3] and [R4] assess trustworthiness in LLMs. However, these benchmarks do not address the efficiency of model compression, which is a critical factor as models continue to grow in size. Although some benchmarks such as [R5] and [R6] benchmark compression methods for LLMs, they only focus on quantization methods.
>
> Different from these methods, our LLMCBench provides a comprehensive evaluation of compression techniques including both sparsity and quantization methods. Moreover, the works [R5] and [R6] mainly focus on the performance of quantized LLM or the effect of specific quantization techniques. The generalization ability, training consumption, inference consumption, and hardware acceleration in our LLMCBench are not well considered in existing benchmark works. We will discuss the differences between our LLMCBench and existing benchmark works in our final version.
>
> [R2] Zhang, Wenxuan, et al. "M3exam: A multilingual, multimodal, multilevel benchmark for examining large language models." *Advances in Neural Information Processing Systems* 36 (2023): 5484-5505.
>
> [R3] Yang, Liu, et al. "Trustworthy LLMs: A survey and guideline for evaluating large language models' alignment." *arXiv preprint arXiv:2308.05374* (2023).
>
> [R4] Lichao, Sun, et al. "Trustllm: Trustworthiness in large language models." *arXiv preprint arXiv:2401.05561* (2024).
>
> [R5] Shiyao, Li, et al. "Evaluating quantized large language models." *arXiv preprint arXiv:2402.18158* (2024).
>
> [R6] Ruihao, Gong, et al. "LLM-QBench: A Benchmark Towards the Best Practice for Post-training Quantization of Large Language Models." *arXiv preprint arXiv:2405.06001* (2024).

---

> ### Author Rebuttal · Authors · 2024-08-16
>
> ****Q4:**** In your hardware acceleration results, why is semi-structured sparsity significantly accelerated in TensorRT-LLM, but not in the other two deployment platforms?
>
> ****A4:**** Sorry for the confusion. The semi-structured sparsity used in Wanda and SparseGPT requires specific implementation from the framework to support the practical acceleration. TensorRT-LLM can support this sparsity pattern, allowing it to achieve practical speedups. In contrast, the other two deployment platforms (vLLM and MLC-LLM) do not support semi-structured sparsity patterns. We will clarify this in our final version.

---

> ### Author Rebuttal · Authors · 2024-08-16
>
> ****Q3:**** In Table 3, can the semi-structured sparsity of Wanda and SparseGPT lead to #MAC reduction?
>
> ****A3:**** Sorry for the confusion. The semi-structured sparsity (2:4 sparsity) sets two weights as zero in each four-weights block. Therefore, it can reduce MACs by skipping the computation of the zero values, which is also demonstrated by the #MACs in Table 3. We will clarify this in our final version.

---

> ### Author Rebuttal · Authors · 2024-08-16
>
> ****Q2:**** I think the author should provide more experimental details, such as the number of shots used in the dataset, the calculation method of inference speed, etc.
>
> ****A2:**** Thanks for your suggestion! We use zero-shot evaluation on all the tasks. For inference speed calculation, we use one A800 GPU to conduct the experiment. We use 128 tokens as both input and output length and calculate the average number of tokens generated per second. We will add these details in our final version.

---

> ### Author Rebuttal · Authors · 2024-08-16
>
> We thank you for your professional and insightful comments. Our response to your comments is as below:
>
> ****Q1:**** This article only focuses on open source models and does not consider closed source models of the GPT series.
>
> ****A1:**** Thanks for your suggestion! We need to use different compression methods to compress the pretrained LLM in our LLMCBench. As a result, we cannot perform the experiments on closed source models. We will clarify this in our final version.

---

### Official Review · Reviewer_7KTd · 2024-07-21

**Rating:** 7
**Confidence:** 3
**Correctness:** The design of the six tracks and the …

**Review:**

The performance of compressed LLMs after deployment is the focus of this work, which is of practical significance. The comprehensive nature of the LLMCBench is commendable, which covers various aspects from performance to trustworthiness. It provides clear insights and suggestions based on the evaluation, which can significantly contribute to the field. However, it would be beneficial to include a discussion on the societal impacts of deploying compressed LLMs, especially regarding trustworthiness and  potential biases.

**Strengths:**

1. The LLM compression methods selected in this paper are all popular and representative pruning or quantization methods, making their results more solid.
2. The paper is neatly structured, with detailed descriptions and clear figures, making it easy for readers to read.
3. By focusing on practical deployment scenarios and using real-world datasets, the benchmark provides valuable insights for real-world applications.

**Additional Feedback:**

There is a question for the author:

- In Figure 1, for Wanda and SparseGPT, did you use the performance of semi-structured pruning or the performance of unstructured pruning to draw the graph?

**Clarity:**

The paper is well-written and easy to follow. I like Figures 1 and 3 of this article, which clearly show the experimental results.

**Documentation:**

There is no open source code link.

**Ethics:**

I do not suspect that.

**Opportunities For Improvement:**

1. In Track 1, "inference ability" is a broad term, encompassing mathematical, commonsense, and more forms of reasoning. I suggest that the authors add some experiments on mathematical tasks.
2. In your conclusions you mentioned some future directions, and I'm wondering what specific types of compression methods you'd like to add to your benchmark?
3. I think you should pay more attention to showing the proposed OM series indicators in the pictures.

**Relation To Prior Work:**

This article will benefit from describing how it differs from BiBench [1], as some of the formulas and graphs are similar.

References:

[1] Qin, Haotong, et al. "Bibench: Benchmarking and analyzing network binarization." *International Conference on Machine Learning*. PMLR, 2023.

**Summary And Contributions:**

In this work, the authors introduce LLMCBench, a comprehensive benchmark designed to evaluate the performance of Large Language Model (LLM) compression techniques. The authors address the need for an effective and fair comparison framework of LLM compression methods. LLMCBench provides a systematic approach to assess compression algorithms across various metrics, including compression performance, generalization ability, training and inference consumption, hardware acceleration, and model trustworthiness.

---

> ### Author Rebuttal · Authors · 2024-08-16
>
> **Q6:** In Figure 1, for Wanda and SparseGPT, did you use the performance of semi-structured pruning or the performance of unstructured pruning to draw the graph?
>
> **A6:** Sorry for the confusion. We use the unstructured pruning results of Wanda and SparseGPT in Figure 1. We will add the explanation in our final version.

---

> ### Author Rebuttal · Authors · 2024-08-16
>
> ****Q5:**** This article will benefit from describing how it differs from BiBench, as some of the formulas and graphs are similar.
>
> ****A5:**** Thanks for your suggestion! BiBench mainly evaluates the binarization algorithm, which does not include LLM compression algorithms. In contrast, our LLMCBench focuses on LLM compression methods, which has different evaluation scopes. Therefore, the analysis on the specific ability of LLM like knowledge or inference ability in our LLMCBench is not considered in BiBench as well. We will discuss the difference in our final version.

---

> ### Author Rebuttal · Authors · 2024-08-16
>
> ****Q4:**** I think you should pay more attention to showing the proposed OM series indicators in the pictures.
>
> ****A4:**** Thanks for your suggestion! We will revise our figures in the final version.

---

> ### Author Rebuttal · Authors · 2024-08-16
>
> ****Q3:**** In your conclusions you mentioned some future directions, and I'm wondering what specific types of compression methods you'd like to add to your benchmark?
>
> ****A3:**** Thanks for your feedback. In our future work, we plan to incorporate knowledge distillation methods like MiniLLM [R1] into LLMCBench as knowledge distillation is another commonly used large model compression method in this area.
>
> [R1] MiniLLM: Knowledge Distillation of Large Language Models, ICLR2024

---

> ### Author Rebuttal · Authors · 2024-08-16
>
> ****Q2:**** In Track 1, "inference ability" is a broad term, encompassing mathematical, commonsense, and more forms of reasoning. I suggest that the authors add some experiments on mathematical tasks.
>
> ****A2:****  Thanks for your suggestion! In our LLMCBench, the "inference ability" mainly examines the reasoning ability of LLM based on its knowledge. Therefore, we selected several mainstream common sense reasoning datasets such as PIQA and HellaSwag in our evaluation. We also follow your suggestion and conduct experiments on GSM8K for mathematical tasks, and the results are shown in the Table below.
>
> | Models | Dense/FP16 | LLM-Pruner | Wanda 50% | Wanda 2:4 | SparseGPT 50% | SparseGPT 2:4 | GPTQ | AWQ | OmniQuant | SmoothQuant |
> |------|----------|----------|---------|---------|-------------|-------------|----|---|---------|-----------|
> | LLaMA2-7B |  14.56     |    1.21    |   4.85    |    1.21   |     6.52      |   1.90        | 13.95| 13.50| 14.02     |   13.95     |
> | LLaMA3-8B |  48.60     |    2.73    |   13.57   |    1.29   |     20.47     |   5.76        | 48.29| 44.88| 46.61     |   45.49     |
>
> The results show that quantization methods can outperform sparsity approaches by a large margin for the mathematical task. We hypothesize that mathematical tasks are more difficult, so it is relatively hard to sparsify the model. The results indicate it is more beneficial to use quantization approaches if we focus on the mathematical task. We will add the experimental results and discuss the observation in our final version.

---

> ### Author Rebuttal · Authors · 2024-08-16
>
> We thank you for your professional and insightful comments. Our response to your comments is as below:
>
> ****Q1:**** It would be beneficial to include a discussion on the societal impacts of deploying compressed LLMs, especially regarding trustworthiness and potential biases.
>
> ****A1:****  Thanks for your suggestion! Our LLMCBench aims to evaluate LLM compression algorithms for practical usage, so it does not have negative social impacts. We have discussed the social impact in our main paper (L309-310). However, considering the compressed LLMs have decreased trustworthiness, they may generate misleading information to users. We will discuss the social impact further in our final version.

---

> > ### Comment · Reviewer_7KTd · 2024-08-24
> >
> > Thanks for your careful rebuttal. My concerns are well addressed in this rebuttal. Thus, I keep my original score to accept this paper.

---

### Author Rebuttal · Authors · 2024-08-16

****General Response****

We sincerely thank AC for handling our manuscript and thank the reviewers for their valuable and insightful feedback:  “systematic approach” and “valuable insights” (Reviewer 7KTd), “nice benchmark” and “great contribution” (Reviewer bNyA), “highly innovative” and “thorough analysis” (Reviewer jkFp), “well-written”, “well-organized” and “well structured” (Reviewer 7KTd, bNyA, jkFp).

In the rebuttal, we have addressed the following concerns:

1. We conduct more experiments according to the reviewers' comments, such as GSM8K and ANLI. (7KTd, jkFp)

2. We explain more specific experimental configurations. (7KTd, bNyA, jkFp)

3. We discuss the possible future work of our LLMCBench. (7KTd)

4. We explain more difference between LLMCBench and other benchmarks in detail.  (7KTd, bNyA)

5. We revise figures according to the reviewers' comments for better presentation. (7KTd, jkFp)

6. We add more detailed explanations of the hardware acceleration results of the three platforms. (bNyA)

7. We include a more detailed analysis on the types of LLM architectures covered by LLMCBench. (jkFp)

We will revise our manuscript accordingly and include these revisions in our final version.

---

### Decision · Program_Chairs · 2024-09-26

**Decision:**

Accept (Spotlight)

**Comment:**

LLMCBench provides a systematic approach to assess compression algorithms across various metrics, including compression performance, generalization ability, training and inference consumption, hardware acceleration, and model trustworthiness.  It tested 18 LLMs, including LLaMA, ChatGLM, etc., covering the current mainstream LLM architectures. The LLMs evaluated range from the smallest 1.3B to the largest 70B, which also covers most open-source LLMs used in the research community. This is a good contribution to the LLM research community.  In the future, it'll be helpful to include the evaluations of recurrent/hierarchical models, such as Mamba, RMT, and HMT.